# Up-to-Date Procedures in Female Stress Urinary Incontinence Surgery: A Concise Review on Bulking Agents Procedures

**DOI:** 10.3390/medicina58060775

**Published:** 2022-06-08

**Authors:** Maurizio Serati, Andrea Braga, Stefano Salvatore, Marco Torella, Maria Carmela Di Dedda, Chiara Scancarello, Chiara Cimmino, Andrea De Rosa, Matteo Frigerio, Massimo Candiani, Alessandro Ferdinando Ruffolo

**Affiliations:** 1Department of Obstetrics and Gynecology, Del Ponte Hospital, University of Insubria, 21100 Varese, Italy; chiarascanca@gmail.com (C.S.); chia.cimmino@gmail.com (C.C.); andreaderosa.unisi@gmail.com (A.D.R.); 2Department of Obstetrics and Gynecology, EOC—Beata Vergine Hospital, 6850 Mendrisio, Switzerland; andrea.braga@eoc.ch; 3Faculty of Biomedical Sciences, Università della Svizzera Italiana, 6900 Lugano, Switzerland; 4Obstetrics and Gynecology Unit, IRCCS San Raffaele Hospital, Vita-Salute University, 20132 Milano, Italy; salvatore.stefano@hsr.it (S.S.); candiani.massimo@hsr.it (M.C.); alesruffolo@gmail.com (A.F.R.); 5Department of Obstetrics and Gynecology, Second Faculty, 80129 Naples, Italy; marcotorella@iol.it; 6Department of Obstetrics and Gynecology, ASST FBF-SACCO Macedonio Melloni Hospital, 20129 Milano, Italy; maria.didedda@asst-fbf-sacco.it; 7ASST Monza, Ospedale San Gerardo, 20900 Monza, Italy; frigerio86@gmail.com

**Keywords:** stress urinary incontinence, urinary incontinence, SUI, bulking agents, urethral bulking, UBA, mini-invasive

## Abstract

To avoid complications related to mid-urethral slings (MUS), alternative procedures to treat stress urinary incontinence (SUI), such as urethral bulking agents (UBAs) have been adopted. The aim of this review is to narratively report the efficacy and safety of UBAs for SUI treatment. For this review, research from PubMed and EMBASE was performed to evaluate relevant studies that were undertaken from January 2012 to January 2022. Nineteen prospective studies were included. Several definitions of subjective and objective success were adopted. At a follow-up of <24 months, significant improvement was widely observed, even if with a heterogeneous rate of success between 32.7–90%, and a reinjection rate of 8.3–77.3%. Compared with other procedures, MUS resulted as significantly superior to UBAs but was balanced by a higher complication rate. Acute urinary retention, urinary tract infection and de novo urgency, and other complications, such as injection site rupture, urethral erosion and particle migration have been described after UBAs. SUI after UBAs treatment resulted in improvements in all studies and can be considered a safe and effective option to treat SUI. However, homogenous and longer-term data lack, limiting general recommendations. Thus, larger RCTs evaluating long-term effects are required.

## 1. Introduction

Currently, mid-urethral slings (MUS) are considered the gold standard of treatment [1] for stress urinary incontinence (SUI). This consideration is sustained by a wide population-based cohort analysis from the National Health Service in England that evidenced in more than 90,000 women receiving MUS a low removal rate of 3.3% and a low failure rate of 4.5% after 9 years of follow-up [2]. 

However, controversy and debates have been raised regarding complications related to surgical anti-incontinence procedures, leading to several court actions in the USA, Canada, UK and some European countries against the use of vaginal slings and tapes [3,4].

In order to avoid surgical techniques and their complications, alternative anti-incontinence procedures have been evaluated, with a renewed interest in the role of urethral bulking agents (UBAs) as a minimally invasive alternative in the management of SUI [5]. 

The principal advantages of the UBAs procedure is the possibility of performing the procedure in the office under local anesthesia, and the possibility of proposing these procedures to patients with severe comorbidities. They also present a reduced rate of adverse events. 

The procedure consists of injections of the agent into the submucosal tissues of the urethra to increase the coaptation of the urethral walls, leading to increased urethral resistance and improved continence. UBAs may be injected trans-urethrally or para-urethrally under urethroscopic control, or alternatively using a purpose-made device (implacer).

Principal injection sites include the bladder neck, between 0.5 and 2 cm distal to the bladder neck and mid-urethra.

Among UBAs, the glutaraldehyde cross-linked bovine collagen (Contigen^®^) was the first agent to be approved by the Food and Drug Administration (FDA) [6]. However, Contigen^®^ and other agents, such as the polytetrafluoroethylene (Teflon^®^), have been discontinued due to immunogenic adverse events such as skin reaction and arthralgia [6]. Currently available UBAs include polyacrylamide hydrogel (Bulkamid^®^); polydimethylsiloxane (Macroplastique^®^); carbon-coated zirconium oxide (Durasphere^®^); calcium hydroxylapatite (Coaptite^®^); and a polydimethylsiloxane silicone gel that polymerizes at the time of injection (Urolastic^®^). 

Even if the last Cochrane Systematic Review [6] stated that “The available evidence base remains insufficient to guide practice”, the last version of the International Consultation on Incontinence recommendations included bulking agents in the group of the first-line surgical treatments for SUI [7]. 

Moreover, one of the last metanalyses showed how the subjective SUI recurrence rate after the UBAs procedure was similar to the perceived recurrence after MUS procedure [8]. 

Hence, our up-to-date review aims to narratively evaluate the efficacy and safety of urethral bulking for SUI treatment. 

## 2. Literature Research

For this narrative review on the efficacy and safety of UBAs for the treatment of SUI, a search of the literature from EMBASE and PubMed was performed, in order to include relevant articles.

The following key words were used for the research: “stress urinary incontinence”; “urinary incontinence”; “incontinence”; “bulking agents”; “bulking”; “polyacrylamide”; “bulkamid”; “polydimethylsiloxane”; “macroplastique”; “calcium hydroxilapatite”; “coaptite”; “polydimethylsiloxane”; and “urolastic”.

The types of study that were included were prospective randomized or non-randomized clinical trials. Retrospective studies, pilot studies, systematic reviews, metanalyses, case series, case reports, comments, letters, animal studies, pre-clinical and basic research experiments were excluded. The search was limited to English language articles that were published in the last 10 years (from January 2012 to January 2022).

The evaluated information was the type of article, sample size, type of UBAs, type of urinary incontinence, first- or second-line therapy, subjective and objective outcomes, reinjection rate, and adverse events.

The patients that were included were women affected by SUI based on the diagnostic criteria of the International Continence Society.

The primary outcomes were to describe the post-operative subjective and objective outcomes in women who received urethral injection of bulking agents. A secondary outcome was to report adverse events. 

## 3. Characteristics of the Included Studies

A total of 596 studies were selected for this review. According to the inclusion and exclusion criteria, another 544 studies were excluded. Further, 33 articles were reviewed and excluded because they did not satisfy the eligibility criteria. Nineteen studies concerning the efficacy and safety of UBAs for the treatment of SUI were included in this review: two of them were RCTs [9,10], while the other 17 studies [11,12,13,14,15,16,17,18,19,20,21,22,23,24,25,26,27] presented a prospective non-controlled or non-randomized design. 

Sample size among the studies ranged from 20 to 229 patients, for a total of 1332 women evaluated.

Most of the literature included is based on the results from the use of the polyacrilamide hydrogel (Bulkamid^®^) [9,10,11,12,13,14,15,16,17,18] and polydimethylsiloxane silicone gel (Urolastic^®^) [20,21,22,23,24], while a smallnumber of the included articles assessed the efficacy of the polydimethylsiloxane (Macroplastique^®^) [19] and the calcium hydroxylapatite (Coaptite^®^) [25] for the treatment of SUI. 

The preferred injection technique was transurethral, while a periurethral approach was rarely adopted. Most of the time, the injection sites were the bladder neck or 0.5 to 2 cm distal to the bladder neck with the mid-urethra minorly involved. The volume of agent that was injected for different types of UBAs varied among studies. 

When the typology of urinary incontinence was evaluated, studies less frequently evaluated cohorts of women complaining of only isolated SUI [9,16,17,21,22,23,24]; indeed, a major part of them reported results for treatment of the stress component of patients that were affected by mixed urinary incontinence (MUI) [9,10,11,12,13,14,17,19,24]. Urodynamic assessment of the type of incontinence was rarely performed before the procedure [16,17,19,23,24]. 

The results were frequently evaluated in a range of 6- to 12-month follow-ups. Rarely, the last follow-up exceeded 24 months, with the longest follow ups reached at 56 and 96 months [15,19].

## 4. Bulking Agents Outcomes 

Several definitions of subjective success and different validated questionnaires were adopted among studies to assess the efficacy of urethral bulking therapies. Eight studies [9,11,12,17,18,19,21,22] subjectively assessed patient’s satisfaction related to improvement in urinary symptoms on a 0–10 or 0–100 VAS. The International Consultation on Incontinence Questionnaire (ICIQ) was adopted in five studies [10,16,18,19,26]. Moreover, four studies evaluated the subjective success of bulking procedures through the Stamey Incontinence Score [21,22,23,24], four studies adopted the PGI-I score [16,17,20,26], one study used the KHQ [15] and two studies used the UDI-6 [14,26]. The scores assessed a significant improvement in urinary incontinence post-operatively in all included studies.

At a follow-up between 3 and 24 months, the subjective success rate among the studies resulted between 32.7% and 90%. When a follow-up longer than 24 months was evaluated, the reported rate of success for the procedure resulted at 42% (96-month follow-up) [15] and 49% (at a mean of 58 months follow-up) [19].

The two most adopted methods to objectively assess changes in SUI symptoms at follow-up were the cough stress test and the 24 h-pad test; a significant improvement was reported in all studies. At a follow-up between 3 and 24 months the objective success rate after the UBAs procedure resulted between 32.7 and 83.6%, with the longest follow-up study reporting a 49% of success at 58 months [19].

General reinjection rate among the studies widely varied between 8.3% and 77.3%.

## 5. Bulking Agents for Primary Stress Urinary Incontinence

In selected cases (i.e., patients with comorbidities, high anesthetic risk, in those who prefer a less invasive approach, etc.) UBAs may be considered a first-line surgical choice. Five studies [9,15,16,17,18] that evaluated the efficacy of UBAs only in women who never previously received other anti-incontinence procedures were included in this review. All the included studies reported results of the polyacrylamide hydrogel injection. The definition of success extremely varies among studies, leading to a wide variability of reported success rate. Subjectively, success rate resulted between 59.8% and 89.7%, while the objective success was reported only in one study resulting at 79.3% [16] at a short follow-up of 12–24 months. At a follow-up longer than 24 months (96 months) the reported subjective success rate resulted at 42% [15]. The reinjection rate after the first bulking procedure resulted between 8.3% and 43%.

## 6. Bulking Agents for Recurrent Stress Urinary Incontinence

UBAs are rarely used in clinical practice as a first line option for SUI treatment because of insufficient evidence [6] and its adoption has been suggested principally in women affected by recurrent SUI. However, few data exist on the role of UBAs for the exclusive treatment of recurrent SUI. Indeed, three studies [13,21,26] that evaluated only women affected by recurrent SUI after a previous anti-incontinence procedure presented the criteria to be included in the current review. 

At a follow-up of 12 to 36 months, the subjective success rate was reported in two studies resulting at 32.7% [21] and 83% [26], while the objective success rate reported was between 32.7% and 83.6%. The reported reinjection rate resulted between 10.6% and 22.4%.

## 7. Bulking Agents vs. Other Surgical Procedures 

According to International Continence Society 2020–2021 Standards [28], complications after the use of meshes, tapes, and grafts in female pelvic floor surgery may involve local complications, complications that affect surrounding organs, and systemic complications. 

During the last 10 years, two clinical studies have compared the subjective and objective outcomes of women treated by urethral bulking and conventional surgical management. In 2018, Wasenda et al. [27] observed a significant increase in the dynamic maximum urethral closure pressure (after cough) in the MUS cohort compared to the UBAs cohort. Moreover, in 2020, Itkonen Freitas et al. [9], using the Urogenital Distress Inventory (UDI-6) and the Incontinence Impact Questionnaire, Short Form (IIQ-7) prospectively showed a significantly higher improvement in quality of life in women who were treated with tension-free vaginal tape (TVT) in comparison with the Bulkamid^®^ procedure. However, only the study by Itkonen Freitas reported a comparison of adverse events with most perioperative complications, including those in 19 tension-free vaginal tape cases (six hematoma, six bladder perforation and seven acute urinary retention) vs. three polyacrylamide hydrogel cases of acute urinary retention (difference 16.0%, 95% CI 7.8–24.9), and all six reoperations due to complications (difference 5.9%, 95% CI 1.2–12.4) were associated with TVT.

Therefore, in all included studies, other surgical procedures resulted as significantly superior to the UBAs treatment, even if they were balanced by adverse events related to MUS (TVT). However, general considerations are limited due to the low number of evaluated clinical studies and to the absence of standardized measures and the consequently large heterogeneity among evaluated outcomes.

## 8. Adverse Events

Even if adverse events, such as acute urinary retention, urinary tract infection and de novo urgency are commonly observed after the surgical treatment of SUI (MUS), these complications have also been observed in several studies assessing the efficacy and safety of UBAs. Indeed, among the included studies, 11 of these reported acute urinary retention with an incidence rate after UBAs procedure ranging between 0% and 10%, with the exception of two Urolastic^®^ studies [20,24,25] that reported a higher incidence of 15–40%. Moreover, urinary tract infection was reported in nine studies with a rate between 2.9% and 11.7%. Seven studies described a post-operative de novo urgency between 0.9% and a maximum of 20% reported in a Urolastic^®^ study [23,24].

Typical complications that are described after UBAs procedures are injection site rupture, urethral erosion and particle migration. Rupture of the injection site was reported twice and only among patients treated with Bulkamid^®^, with an incidence rate of 0.7% and 1.7%. Erosion was evaluated in four studies [21,22,23,24], and reported exclusively in Urolastic^®^ studies, ranging from 0% to 5%. None of the studies that were included assessed the particle migrations.

Dyspareunia was a complication reported between 3.8% and 10% of patients after treatment with Urolastic^®^. 

## 9. Limitations of the Included Studies

Despite the selective inclusion of studies with a prospective design, few RCTs have been produced in the last 10 years. Indeed, only one of them aimed to compare UBAs to MUS, and there are no studies that directly compared the synthetic UBAs that are currently adopted in clinical practice.

The large heterogeneity of study populations, SUI diagnosis (urodynamically or not), evaluated outcomes, definition of success and time to follow-up leads to the wide variability in reported success rates, confirming the difficulty of extrapolating recommendations for clinical practice.

Moreover, the wide heterogeneity of indications for UBAs procedures, such as primary vs. secondary line-setting and pure SUI vs. MUI can be considered as confounding factors affecting the outcomes that were reported in each study.

## 10. Conclusions

Despite the heterogeneity of the evaluated outcomes among different studies, there are more data to support UBAs as a feasible and safe technique to treat SUI, in relation to the low complication rate and the significant improvement in urinary symptoms demonstrated across all the included studies. MUS demonstrated to be the most effective solution, even if the latest evidence showed a similar subjective cure rate among procedures. However, the complication rate for MUS resulted as higher than for UBAs procedures. Indeed, counseling for bulking agents should include the minimal risks of the procedure and the cure, or improvement, in terms of continence, and women should be aware of potential repeat injection. Indeed, when a treatment is proposed to a patient it is crucial remember the level of evidence that has been reported. Therefore, objectives of further research should be the assessment of the indication for UBAs as first- or second-line therapy, a comparison of safety and efficacy among different UBAs, and evaluation of longer-term outcomes. Moreover, homogeneity of the evaluated outcomes is desirable in order to better generalize the results of future investigations.

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
