# Peer review of "Up-to-Date Procedures in Female Stress Urinary Incontinence Surgery: A Concise Review on Bulking Agents Procedures"

_medicina, 2022, doi:10.3390/medicina58060775_

Round 1

Reviewer 1 Report

Comments on medicina-1743230

Title: Uptodate In Female Stress Urinary Incontinence Surgery: A Concise Review on Bulking Agents Procedures

The authors conducted the review to demonstrate the role of urethral bulking agents (UBA) in the management of women with stress urinary incontinence (SUI). Until now, the aforementioned approach needs more solid data to support its efficacy and/or effectiveness, since mid-urethral sling may be considered as the most effective therapeutic choice for treatment for SUI. In terms of considering the complication of slings, authors would like to evaluate the alternative treatment, UBA, to test its effectiveness. Authors did their efforts to perform the literature review between January 2012 and January 2022 and they found a total of 19 prospective studies.  Since the studies are heterogenous with great variation of success rates, ranged from 32.7% to 90%. Additionally, many complications are also found, including urinary tract infection, de novo urgency, and acute urinary retention. Furthermore, the long-term effect is sill scare. Some recommendations are shown below.

1.     In this review paper, the strategy to obtain this review should be included. For example, flow diagram including identification, screening, eligibility and inclusive criteria had better be provided. Authors should also describe the exclusion criteria in the diagram.

2.     Since there are many similar studies in hand to evaluate the efficacy of UBA (ex : PMID: 29934769) with the similar conclusion, to show the better outcomes and quality of the text, the meta-analysis and draw the forest plot for success rate, complication and subjective improvement could be considered.

Author Response

The authors conducted the review to demonstrate the role of urethral bulking agents (UBA) in the management of women with stress urinary incontinence (SUI). Until now, the aforementioned approach needs more solid data to support its efficacy and/or effectiveness, since mid-urethral sling may be considered as the most effective therapeutic choice for treatment for SUI. In terms of considering the complication of slings, authors would like to evaluate the alternative treatment, UBA, to test its effectiveness. Authors did their efforts to perform the literature review between January 2012 and January 2022 and they found a total of 19 prospective studies.  Since the studies are heterogenous with great variation of success rates, ranged from 32.7% to 90%. Additionally, many complications are also found, including urinary tract infection, de novo urgency, and acute urinary retention. Furthermore, the long-term effect is sill scare. Some recommendations are shown below.

First of all, thank you for your time spent to precisely review our manuscript.

  1. In this review paper, the strategy to obtain this review should be included. For example, flow diagram including identification, screening, eligibility and inclusive criteria had better be provided. Authors should also describe the exclusion criteria in the diagram.

The Medicina Journal asked us to narrativelyreport the role of UBAs in SUI treatment. That’s why we didn’t find appropriate to insert a real flow diagram as we use to in systematic reviews. However, we decided to report in the Results the number of articles that have been evaluated and excluded in relation to the inclusion and exclusion criteria (key words for research, type of article selected, publication period, type of patient evaluated).

  1. Since there are many similar studies in hand to evaluate the efficacy of UBA (ex : PMID: 29934769) with the similar conclusion, to show the better outcomes and quality of the text, the meta-analysis and draw the forest plot for success rate, complication and subjective improvement could be considered.

Even if meta-analysis is desirable to better understand the efficacy and safety of UBAs in SUI treatment, as already mentioned in the previous answer the Journal asked us to narrativelyreport the role of UBAs. In our opinion, it is not methodologically possible and correct to perform a meta-analysis on results coming from a narrative review.

Reviewer 2 Report

Collection of the data is fair enough. Limitation of the studies is emphasised, and heterogeneity is mentioned. However, for someone not directly included in the treatment of SUI, the conclusion might be misleading. More emphasise should be put in responsibility of the physician to correctly and comprehensively explain all the benefits and weakness of different procedures. Namely, main goal of reviews and analyses of this type is to inform and give proper recommendations to end-users. 

Author Response

Collection of the data is fair enough. Limitation of the studies is emphasised, and heterogeneity is mentioned. However, for someone not directly included in the treatment of SUI, the conclusion might be misleading. More emphasise should be put in responsibility of the physician to correctly and comprehensively explain all the benefits and weakness of different procedures. Namely, main goal of reviews and analyses of this type is to inform and give proper recommendations to end-users.

Thank you for your time and for your suggestion. We tried to better emphasize responsibility of physician in the Conclusion as suggested.

Round 2

Reviewer 1 Report

Congratulation and I totally agree with your response.